# Understanding and mitigating the impact of ambient mRNA contamination in single-cell RNA-sequencing analysis

Jantarika Kumar Arora[1,2]*, Louisa K. James[1], Varodom Charoensawan[2,3,4,5,6,7]*

1 Centre for Immunobiology, Faculty of Medicine and Dentistry, Blizard Institute, Queen Mary University of London, London, England, 2 Department of Biochemistry, Faculty of Science, Mahidol University, Bangkok, Thailand, 3 Department of Biochemistry, Faculty of Medicine Siriraj Hospital, Mahidol University, Bangkok, Thailand, 4 Division of Medical Bioinformatics, Research Department, Faculty of Medicine Siriraj Hospital, Mahidol University, Bangkok, Thailand, 5 Siriraj Genomics, Faculty of Medicine Siriraj Hospital, Mahidol University, Bangkok, Thailand, 6 Integrative Computational BioScience (ICBS) Center, Mahidol University, Nakhon Pathom, Thailand, 7 School of Chemistry, Institute of Science, Suranaree University of Technology, Nakhon, Ratchasima, Thailand

* j.kumararora@qmul.ac.uk (JKA); varodom.cha@mahidol.ac.th (VC)

## Abstract

Droplet-based single-cell RNA sequencing (scRNA-seq) frequently encounters significant challenges from contamination of cell-free mRNAs, known as "ambient mRNAs", which can substantially distort single-cell transcriptome data interpretation to a large extent. In this study, we investigate the impact of ambient mRNA contamination on differential gene expression and biological pathway enrichment analyses, using two independent scRNA-seq datasets: ten peripheral blood mononuclear cells (PBMCs) samples from dengue-infected patients and forty-two scRNA-seq samples of human fetal liver tissues. We apply two independent ambient mRNA correction approaches – CellBender (automate correction) and SoupX (using a predefined set of potential ambient mRNA genes). We demonstrate that ambient mRNA transcripts appear among differentially expressed genes (DEGs), subsequently leading to the identification of significant ambient-related biological pathways in unexpected cell subpopulations before ambient mRNA contamination correction. In contrast, after suitable correction, we observe a reduction in ambient mRNA expression levels, resulting in improved DEG identification and leading to the highlight of biologically relevant pathways specific to cell subpopulations. Our study underscores the importance of understanding and applying appropriate corrections for ambient mRNA contamination to enhance the reliability and accuracy of scRNA-seq data analyses, thereby improving the robustness of data interpretation in droplet-based scRNA-seq datasets.

**Data availability statement:** Single-cell RNA-sequencing datasets: The raw sequencing reads of the single-cell RNA-seq datasets used in this study include the peripheral blood mononuclear cell (PBMC) datasets, which consists of eight single-cell experiments from dengue patients and one healthy donor [13], available through the ArrayExpress repository: E-MTAB-9467. Another healthy sample was obtained from the 10x Genomics website (4k PBMCs from a healthy donor, Single Cell Gene Expression Dataset by Cell Ranger 2.1.0, 10x Genomics, 2017, November 8, https://support.10xgenomics.com/single-cell-gene-expression/datasets/2.1.0/pbmc4k). In addition, the raw sequencing data of forty-two single-cell experiments of human fetal liver tissues [21] were obtained from the ArrayExpress database, under the accession number: E-MTAB-7407. The raw and filtered 10x Genomics species-mixing dataset, which contains a mixture of human HEK293T and mouse NIH3T3 cells, is available at https://www.10xgenomics.com/datasets/10-k-1-1-mixture-of-human-hek-293-t-and-mouse-nih-3-t-3-cells-3-v-3-1-3-1-standard-6-0-0, retrieved on 20 June 2025.

**Funding:** This project is funded by the mid-career researcher grant from National Research Council of Thailand (NRCT) and Mahidol University (N42A670557) through VC. The funders had no role in study design, data collection and analysis, decision to publish, or preparation of the manuscript.

**Competing interests:** The authors have declared that no competing interests exist.

## Introduction

Single-cell RNA-sequencing (scRNA-seq) has become a powerful technique for investigating transcriptomic profiles and complex cellular heterogeneity at the single-cell resolution [1–3]. This technology offers not only profound insights into cellular heterogeneity, but also improves our understanding on the functions of highly complex biological systems in both normal and disease-related physiological contexts [4,5]. Droplet-based scRNA-seq platforms, such as Drop-seq [6], inDrop [7], and Chromium 10x Genomics [8], have been widely implemented in various biological contexts, mainly due to their capacity to capture a large number of individual cells at a relatively low cost per cell [9]. These platforms are also suitable for detecting novel cell types [10,11], as well as identifying cell subpopulations within intricate biological samples [12–14].

Despite the several advantages of droplet-based single-cell technologies, one of the important challenges is the contamination of cell-free mRNAs, frequently referred to as "ambient mRNA". This contamination can significantly confound the biological interpretation of single-cell datasets, as demonstrated in previous studies [15–20]. In brain single-nuclei RNA sequencing, for example, previously annotated neuronal cell types were separated by ambient mRNA contamination and immature oligodendrocytes were found to be contaminated with ambient mRNAs [16]. However, after computational removal of this ambient contamination, committed oligodendrocyte progenitor cells (a rare population) were detected, which had not been annotated in most previous adult human brain datasets [16]. This underscores how ambient mRNA contamination can impact cell type annotation. Several computational tools, including SoupX [20], CellBender [17], and DecontX [19] among others, have been developed to estimate and remove ambient mRNAs contamination, subsequently improving the quality of expression matrices and enhancing the expression pattern of cell type-specific marker genes [15,17–20]. Previous studies have also demonstrated the impact of ambient mRNA contamination on downstream analyses [16,18]. However, there are still several aspects of ambient mRNA contaminations that remain to be characterised. Despite these advancements, the effects of ambient mRNA correction on certain downstream analyses, including differential gene expression and pathway enrichment analyses, particularly at the subpopulation levels, remains largely unclear.

In this study, we have performed comprehensive analyses to evaluate the impact of ambient mRNA contamination on the biological interpretation of actual biological scRNA-seq datasets. These included re-analyses of a time-course scRNA-seq dataset of the immune cell responses during the acute phase of dengue infection patients [13], and of an integrated dataset of forty-two scRNA-seq samples obtained from human fetal liver tissues [21], to demonstrate our points. We employed two independent ambient mRNA correction tools, CellBender [17] and SoupX [20]. To ensure a more comprehensive assessment of the impact of ambient mRNA contamination on downstream analyses, independent of methodological differences between correction approaches, we provided a predefined set of predicted potential ambient mRNA genes for SoupX correction, whereas CellBender performed automated prediction and correction. We went on to investigate the influence of contaminated ambient

mRNAs on downstream analyses, focusing on the identification of differentially expressed genes (DEGs) and biological pathway enrichments within T and B cell subpopulations. Comparing transcriptomic profiles of these immune cell subpopulations before and after ambient mRNA correction revealed an improvement in DEG identification, subsequently leading to the emergence of biologically relevant pathways specific to cell subpopulations after correction. Overall, our study highlights the critical importance of addressing ambient mRNA contamination to enhance reliability of scRNA-seq data interpretation and downstream biological insights.

## Materials and methods

### Single-cell RNA-sequencing datasets

The raw sequencing reads of the single-cell RNA-seq datasets used in this study include the peripheral blood mononuclear cell (PBMC) datasets, which consists of eight single-cell experiments from dengue patients and one healthy donor [13], available through the ArrayExpress repository: E-MTAB-9467. Another healthy sample was obtained from the 10x Genomics website (4k PBMCs from a healthy donor, Single Cell Gene Expression Dataset by Cell Ranger 2.1.0, 10x Genomics, 2017, November 8, https://support.10xgenomics.com/single-cell-gene-expression/datasets/2.1.0/pbmc4k). In addition, the raw sequencing data of forty-two single-cell experiments of human fetal liver tissues [21] were obtained from the ArrayExpress database, under the accession number: E-MTAB-7407. The raw and filtered 10x Genomics species-mixing dataset, which contains a mixture of human HEK293T and mouse NIH3T3 cells, is available at https://www.10xgenomics.com/datasets/10-k-1-1-mixture-of-human-hek-293-t-and-mouse-nih-3-t-3-cells-3-v-3-1-3-1-standard-6-0-0, retrieved on 20 June 2025.

### Pre-processing and quality control of scRNA-seq data

**Data pre-processing.** All datasets used in this study were processed using the same consistent pipeline described as follows. Raw FASTQ files were aligned and quantified using the CellRanger Single-Cell Software Suite (version 8.0.1) and the reference human genome GRCh38-2024-A (10x Genomics, USA). Standard preprocessing steps, including normalization of gene expression levels, scaling, clustering, and dimensionality reduction, of individual single-cell data was carried out using Seurat V.5.2.1 [22]. Expression levels of genes in each cell were normalised using the LogNormalize approach available from the *NormalizaData* function, where the unique molecular identifier (UMI) counts of each gene were divided by the total number of UMIs per cell, then multiplied by a scale factor of $10^4$, and subsequently log-transformed. Cell clusters were identified using the *FindClusters* function from the Seurat package (V.5.2.1) [22] with the default settings. Dimensionality reduction was performed using the Uniform Manifold Approximation and Projection (UMAP) method via the *RunUMAP* function, with the first 10 principal components (PCs). Cells with the total mitochondrial gene expression exceeding 10% were excluded. Doublets were detected and discarded using DoubletFinder [23] with the default settings.

**Ambient mRNA correction.** To correct the expression levels of ambient mRNA contaminations, CellBender (v.0.3.0) [17] and SoupX (v1.6.2) [20] were applied using the default settings for all libraries, except for the 10x Genomics species-mixing dataset, where contamination fraction was estimated automatically using *autoEstCont* with the parameters: tfidfMin = 0.01, soupQuantile = 0.8, and forceAccept = TRUE. The raw and filtered gene-barcode matrices were used as inputs to estimate the expression profiles of ambient mRNAs. To enhance the accuracy of estimating the contamination fraction in individual cells, we incorporated a curated set of genes that were not typically expressed by cells of a certain type, along with clustering information. Specifically, for the single-cell dengue dataset, we included a set of immunoglobulins (Ig) genes, whereas for the human fetal liver dataset, a set of hemoglobin (Hb) genes was provided.

### Data integration and normalisation

After quality control steps described above, individual samples were integrated using the *SCTransform* v2 normalisation approach from Seurat V.5.2.1 [22,24], with the default settings. Cell clustering and UMAP dimensionality reduction were

then performed using *FindClusters* and *RunUMAP*, using the first 30 PCs. Expression levels of genes in each cell were normalised using the *NormalizaData* function, as mentioned above.

### Cell type annotation and subpopulation analysis

Cell type annotation was performed using Azimuth (v.0.5.0) [25]. For the dengue datasets, we applied the "Human - PBMC" reference [25], while the "Human-Liver" reference [26–30] was used for fetal tissue datasets. T and B cell populations were selected based on the "celltype.l1" and "celltype.l2" annotation level from Azimuth [25]. Additionally, we exclude non-T and non-B cells using known canonical marker genes (S1 Table) for T cells and the annotations as in the original article [21] for B cells where relevant to enhance the accuracy of downstream analyses at subpopulation levels (Figs 3 and 4).

### Differential gene expression analysis

A total of 38,606 genes were tested for differential gene expression (DEG) analysis using the Wilcoxon rank sum test, implemented in the *FindAllMarkers* function of Seurat V.5.2.1 [22]. Genes were considered differentially expressed using Seurat's default parameters – specifically, they were expressed in at least 10% of cells in either group and had a minimum log2 fold-change threshold of 0.1. DEGs were defined as those with an adjusted p-value < 0.01 after Bonferroni correction for multiple testing (also using default settings). These default parameters were used for the initial identification of DEGs and were not intended to define biologically meaningful expression changes. The total numbers of DEGs passing these cutoffs across datasets are listed in S2 Table.

### Pathway analyses

g:Profiler2 (v.0.2.3) [31] was applied to systematically assess biological processes using human reference genes from GRCh38.p14 and focusing on Gene Ontology Biological Process (GO:BP). To reduce the false positives, the false discovery rate (FDR) approach was applied for multiple testing correction, and adjusted p-values < 0.05 were considered statistically significant. For GO:BP analyses, DEGs were selected by ranking genes based on average log2 fold-change (avg_log2FC), without applying a fixed fold-change threshold. This rank-based approach priorities the most biologically distinct genes and avoids arbitrarily strict cutoffs, which may be too stringent — particularly when comparing results before and after ambient mRNA correction. The top 20 DEGs (ranked by avg_log2FC) were selected in the dengue datasets and for each seurat cluster in the 10x Genomics species-mixing dataset (10x Genomics; https://www.10xgenomics.com/data-sets/10-k-1-1-mixture-of-human-hek-293-t-and-mouse-nih-3-t-3-cells-3-v-3-1-3-1-standard-6-0-0, retrieved on 20 June 2025). In the human fetal tissues, the top 2000 DEGs of "pro B cells" were selected. To visualise unique and overlapping GO:BP terms before and after ambient mRNA correction, Venn diagrams were constructed using InteractiVenn [32].

## Results and discussion

### Evaluating the impact of ambient mRNA contamination in the peripheral blood mononuclear cell samples

To demonstrate the presence of ambient mRNA contamination and its impact on downstream analysis, we first utilised a publicly available peripheral blood mononuclear cell (PBMC) dataset [13]. The dataset exhibited contamination from ambient mRNAs or background noise, as evidenced by the presence of nonzero counts of known marker genes in unexpected cell types [17,19,20] (S1 Fig in S1 File). After performing the quality control steps, we assigned immune cell types based on cell type annotations from Azimuth [25] using the "Human - PBMC" reference [25], supplemented by established marker genes where relevant (Fig 1A, S1 Table, and S2 Fig in S1 File). Major PBMC populations, including CD8 T cells, NK cells, B cells and Plasma cells (PCs), exhibited canonical marker gene expression consistent with their identity (Fig 1A, S1 Table, and S2 Fig in S1 File).

Based on the expression of known canonical marker genes, we observed relatively high expression levels of the T cell markers, *CD3E* (encoding the CD3-epsilon polypeptide to form the T cell receptor complex) and *TRAC* (T cell Receptor Alpha Constant) in annotated T cells, with average expression ranging from 1.29–1.43 and up to 75% of T cells expressing the genes, compared to other clusters (average expression 0.03–0.82, <54% expressing cells) (Fig 1B, S3 Table, and S2 Fig in S1 File). Similarly, B cell lineage genes *MS4A1* (B-lymphocyte-specific membrane protein) and *CD79A* (B-cell antigen receptor complex-associated protein) exhibited higher expression levels (1.26–2.10) and were expressed in > 76.23% of B cells and PCs, compared to other cell types (average expression <0.14 and < 16% expressing cells) (Fig 1B, S4 Table, and S2 Fig in S1 File). In contrast, certain immunoglobulin (Ig) genes (referred to as "B cell-related genes", herein), comprising *IGKC*, *IGHG1*, *IGHG4*, and *JCHAIN*, were expressed across multiple populations at relatively low expression levels (0.45–3.33) and in up to 99% of these cells expressing the genes, when compared to B cells and PCs expected to express these genes (average expression 1.06–6.08 and 62–100% expressing cells) (Fig 1B, S4 Table, and S2 Fig in S1 File).

For droplet-based scRNA-seq technology, a small amount of cell-free mRNA molecules can be distributed into droplets and subsequently sequenced alongside mRNAs from intact single cells [17,19,20], resulting in non-zero molecule counts within droplets containing cell-free mRNAs [20]. The cell-free mRNA molecules, commonly known as "ambient mRNAs", are likely derived from cells that have undergone lysis, stress, or apoptosis during the experiment [17]. This can subsequently introduce several challenges for the analyses, including known marker genes of certain cell types being observed in other unexpected cell types [20].

## Appropriate correction of ambient mRNA levels reduced pervasive expression of marker genes

Having observed substantial levels of ambient mRNA contamination in the selected scRNA-seq dataset (Fig 1, S2–S4 Tables, and S1-S2 Figs in S1 File), we implemented ambient mRNA correction using two independent methods: CellBender [17] and SoupX [20]. These tools estimate the contamination fraction of ambient mRNAs within individual cells and subsequently adjust the transcription levels in expression matrices. CellBender performs automated prediction and correction of ambient mRNA contamination, whereas SoupX is capable of either automated or manual ambient mRNA correction, allowing for fine-turning of the detection process [17,20]. Typically, known marker genes of specific cell types being expressed at relatively low levels across other cell types can frequently be considered ambient mRNA contamination [17,20]. In this particular PBMC dataset, we specified Ig genes as potential contaminating genes for estimating contamination fractions with SoupX. Evidently, Ig genes were indeed identified as potential ambient mRNA contamination among the top 10 genes predicted by SoupX (S5 Table). Moreover, they exhibited relatively low expression levels across multiple cell types (Fig 1, S2–S4 Tables, and S1–S2 Figs in S1 File).

After applying the ambient mRNA correction, we compared gene transcription levels to those before correction (Fig 2; see S3 Fig for SoupX in S1 File). The transcription levels of canonical marker genes in corresponding cell types, including selected T and B cell markers, remained consistently high and relatively unchanged after correction (S6 Table and S4 Fig in S1 File) with both CellBender (S6 Table and S5 Fig in S1 File) and SoupX (S6 Table and S6 Fig in S1 File). Specifically, the average expression levels of these genes across all cells were 0.36 before correction, 0.34 after CellBender and 0.35 after SoupX corrections (S6 Table). This suggests the robustness of the ambient mRNA correction step in accurately preserving the expression patterns of well-established cell type markers.

Conversely, after the correction, we observed significant decreases in the transcription levels of B cell-related genes in "non-B cell" populations following correction with both CellBender and SoupX (Figs 2A-2B, see S3 Fig for SoupX, and S7 Fig in S1 File). The average expression levels of B cell-related genes in "non-B cells" dropped from 0.47 before correction to 0.07 after CellBender and 0.05 after SoupX correction (S7 Table). These results suggest that the correction process effectively removed the expression signals originating from ambient mRNAs. Additionally, we observed a reduction in the percentage of cells expressing these genes after correction (S8 Table). Importantly, the transcription levels of these B

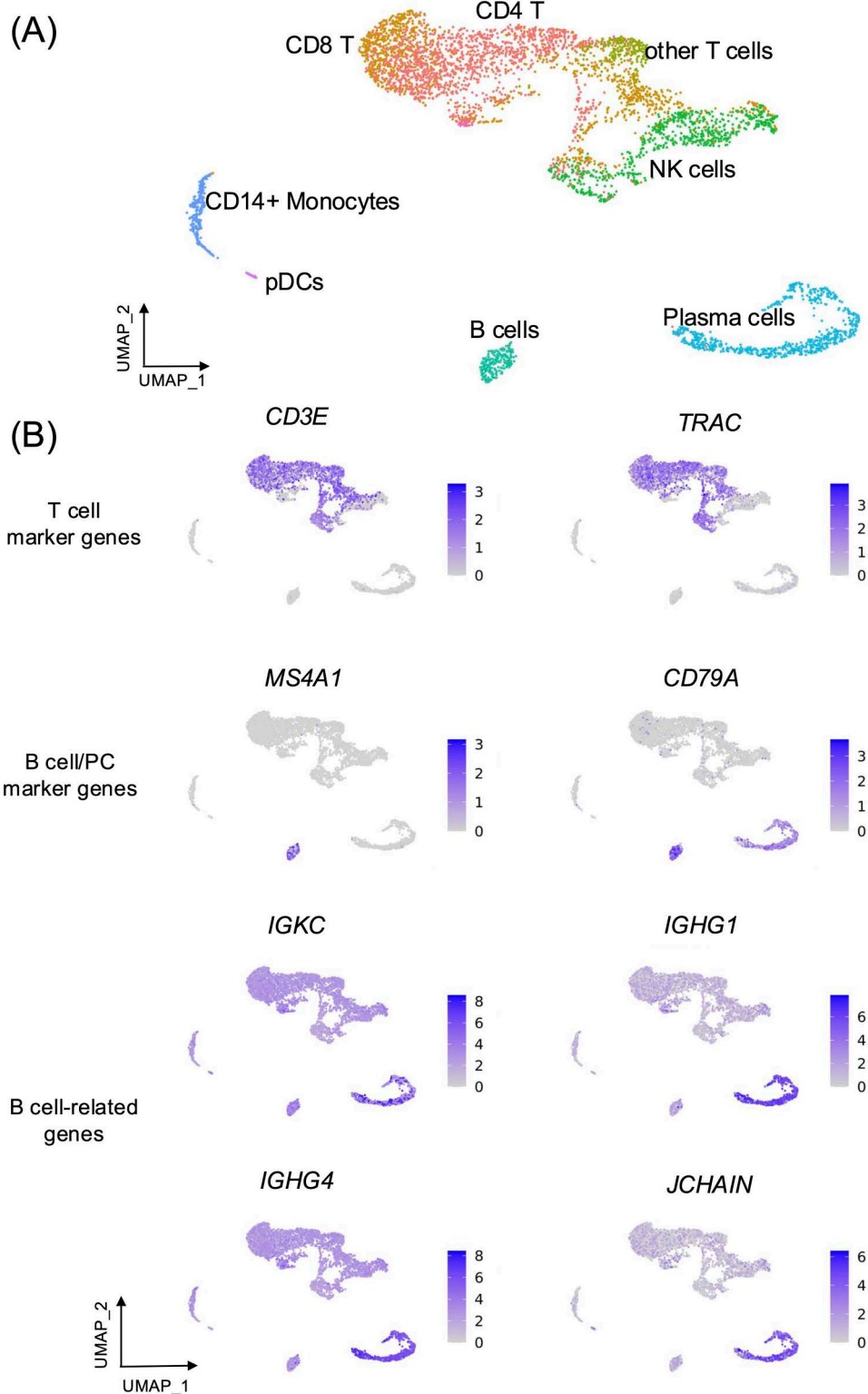

**Fig 1. Expression of B cell-related genes in non-B cell populations.** (A). Uniform Manifold Approximation and Projection (UMAP) plot showing single-cell transcriptome profiles of peripheral blood mononuclear cells (PBMCs) of the dengue dataset [13], colored by cell types annotated using Azimuth, supplemented by canonical cell-type marker genes where relevant (S1 Table and S2 Fig in S1 File). (B). UMAP feature plots demonstrating the

relative transcription levels of T cell marker genes (first panel), B cell and plasma cell (PC) marker genes (second panel), and B cell-related genes (third and fourth panels). Cell types are annotated using Azimuth, supplemented by canonical cell-type marker genes (S1 Table) where relevant. pDCs = plasmacytoid dendritic cells.

cell-related genes in "B cells" (comprising B cells and PCs here) remained unchanged before and after ambient mRNA correction (Fig 2B, see S3 Fig for SoupX in S1 File), indicating that the correction process successfully retained the intrinsic gene expression patterns specific to B cells.

## Enhancement of differential gene expression and biological pathway enrichment in T cell subpopulations after ambient mRNA correction

As we observed the presence of ambient mRNA contamination across T cells in the PBMC dataset used to showcase our points here (Figs 1–2, and S1-S7 Figs in S1 File), we then investigated the extent to which the ambient mRNA correction impacts subsequent bioinformatic analyses, specifically focusing on the identification of differentially expressed genes (DEGs) and biological pathway enrichments. Following the integration of ten scRNA-seq profiles from the same representative study [13], we observed no apparent batch effects in T cells (S8 Fig in S1 File). To specifically assess the impact of ambient mRNA contamination on T cell subpopulations without confounding signals from other immune cell types, we further extracted T cell subpopulations based on the "celltype.l2" annotation level from Azimuth. Non-T cell populations were then excluded using established canonical marker genes (S1 Table and S2 Fig in S1 File). The number of annotated T cells remained relatively comparable before and after ambient mRNA correction (S9 Table and S9 Fig in S1 File), suggesting that contamination did not impact the annotation.

While T cell subpopulations expressed T cell markers, we also observed widespread expression of Ig genes, including *IGKC, IGHA1, IGHG4*, and *IGLC2*, across almost all annotated T cell subsets, despite these genes being specific to B cells and PCs (S10 Fig in S1 File). After applying ambient mRNA correction, the expression of these Ig genes was noticeably reduced across T cells (S11 Fig for CellBender and S12 Fig for SoupX in S1 File), with average expression levels decreasing from 0.54 before correction, to 0.10 after CellBender and 0.06 after SoupX (S10 Table), suggesting effective removal of background contamination. Specifically, following correction using CellBender, which performs automated ambient mRNA removal, *IGKC* and *IGHG4* were still observed but were primarily restricted to "CD4 Proliferating" and "CD8 Proliferating" T cells (S11 Fig in S1 File and S10 Table). In contrast, after correction using SoupX, in which Ig genes were explicitly provided as predefined potential ambient mRNA genes, expression levels of these Ig genes were reduced to low or near-zero counts across T cells (S12 Fig in S1 File and S10 Table).

We further investigated how ambient mRNA contamination influences the identification of DEGs within T cell subpopulations. Without the ambient mRNA correction, fifteen B cell-related genes appeared among the top 20 DEGs in at least one of the three biological conditions (acute, convalescent, or healthy control), when compared to the other two conditions, across a wide range of well-characterised T cell subpopulations (Figs 3A; left panel). After correction, Ig genes were still present but at markedly lower expression levels in most T cell subpopulations compared to before correction, as shown using the same color bar (Fig 3A and S13 Fig in S1 File). The reduction in Ig gene expression levels was observed with both the results from the CellBender and SoupX pipelines but was slightly more pronounced in the SoupX result in this case (S13 Fig in S1 File). Additionally, nine Ig genes remained in the top 20 DEGs after CellBender correction (S13 Fig in S1 File; middle panel) and eleven Ig genes after SoupX (S13 Fig in S1 File; lower panel), out of the fifteen Ig genes previously identified as DEGs (S13 Fig in S1 File; upper panel). Accordingly, six Ig genes – *IGHM, IGHG3, IGHA2, IGLC2, IGLC3,* and *IGKV1–18* – after CellBender correction and four – *IGHM, IGHA2, IGLC3,* and *IGKV1–18* – after SoupX correction, were effectively removed from the top 20 DEGs in T cells subpopulations (S13 Fig in S1 File).

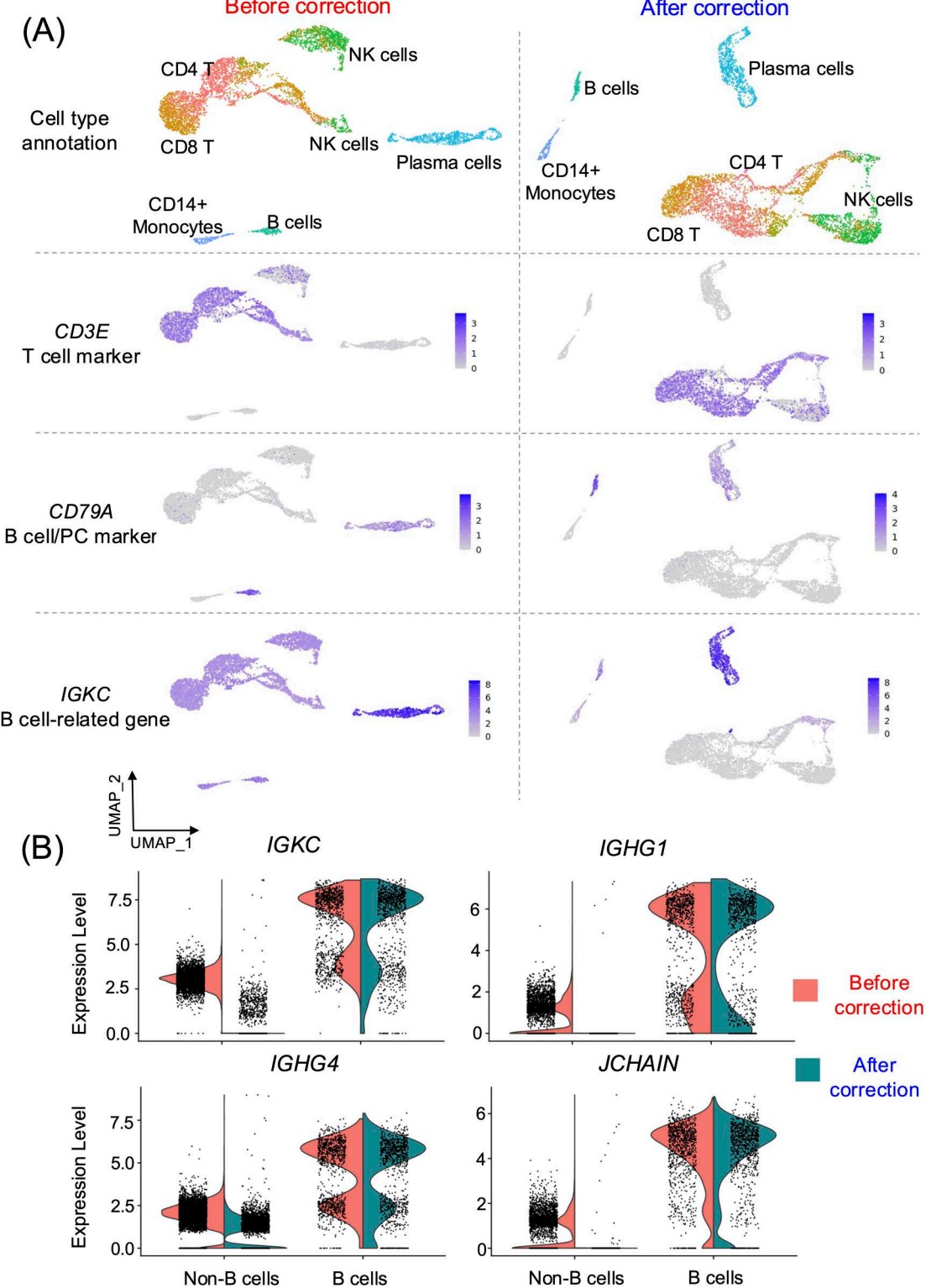

**Fig 2. Expression profiles of ambient mRNAs before and after correction.** (A). UMAP plot representing the single-cell transcriptome profiles of PBMCs before (left panel) and after (right panel) ambient mRNA correction using CellBender [17] (refer to S3 Fig in S1 File for results with SoupX [20]). Cell types are annotated using Azimuth [25] and supplemented with established canonical marker genes where relevant (S1 Table and S2 Fig in

S1 File). (B). Violin plots showing the normalised transcription levels of the B cell-related genes in each of the non-B cell and B cell populations (comprising B cells and PCs), comparing between before and after ambient mRNA correction using CellBender (see S3 Fig for SoupX in S1 File).

We next explored the impact of ambient mRNA contamination on pathway enrichment analysis, based on Gene Ontology Biological Processes (GO:BPs) of the top 20 DEGs (ranked by average log2 fold-change) in each biological condition (acute, convalescent, or healthy control) across T cell subpopulations (Fig 3B). While the number of input genes for pathway analysis was similar before and after ambient mRNA correction, we identified 17 significant GO:BP terms prior to correction, all of which were influenced by Ig genes (contributing from 34% to 100%; S11 Table). In contrast, after correction, the number of significant GO:BP terms increased substantially – 191 GO:BPs in the CellBender result (S12 Table and S15 Fig in S1 File), and 46 significant GO:BP terms in the SoupX result (S13 Table and S16 Fig in S1 File; top panel). The low number of significant biological pathways before correction is likely due to the presence of non-specific genes, such as Ig genes from B cells and PCs, which dominate the pathway analysis and hence obscure relevant biological signals. Conversely, ambient mRNA correction adjusted the expression levels and rankings of DEGs, allowing the detection of more biologically relevant genes and yielding a marked increase in statistically significant GO:BP enrichments.

Interestingly, out of the 17 GO:BP terms identified before correction, 13 were uniquely observed prior to ambient mRNA removal, compared to after correction with the CellBender result (Fig 3B; left panel and S15 Fig in S1 File). These 13 GO:BP terms indeed represented significant biological pathways that have been shown to be B-cell specific pathways and upregulated during acute dengue infection [13], including GO:0016064 (immunoglobulin mediated immune response), GO:0019724 (B cell mediated immunity), GO:0006959 (humoral immune response), GO:0002460 (adaptive immune response based on somatic recombination of immune receptors built from immunoglobulin superfamily domains), and GO:0050853 (B cell receptor signaling pathway) (Fig 3B; left panel). However, this was not the case after the correction using SoupX, where all 17 GO:BPs were identified. Several GO terms, including the B-cell specific pathways mentioned above, showed lower statistical significance (S16 Fig in S1 File). This suggests that the correction effectively reduced the impact of ambient mRNA contamination, which likely contributed to inflated statistical significance of certain pathways before correction.

We then observed four GO:BP terms – GO:0050896 (response to stimulus), GO:0098542 (defense response to other organism), GO:0006955 (immune response), and GO:0042742 (defense response to bacterium) – that were identified both before and after the CellBender correction (S15 Fig in S1 File). Remarkably, while these GO:BP terms remained consistent, the composition of the intersection genes contributing to them changed significantly (S15 Fig in S1 File). Specifically, before correction, these pathways were dominated by Ig genes, likely due to ambient mRNA contamination originating from B cells and PCs. After the correction using CellBender, however, these pathways were instead associated with contributions from interferon-stimulated genes (ISGs) and interferon-induced gene products (e.g., *ISG15, IFI44L, IFI27, IFIT2, IFIT3*), which were previously undetected (S15 Fig in S1 File). Notably, these genes have been reported to be highly expressed in specific T cell subsets during acute dengue infection [33,34].

Among the 187 uniquely identified GO:BPs after CellBender correction, we observed biological pathways associated with cell cycle processes, cell division, and DNA repair (Fig 3B), which have been identified in HLA-DR+ CD38+ CD8 T cells during dengue fever [35]. Additionally, we found type I interferon-related pathways (Fig 3B), which have been shown to be expressed in T cells [13,34,36]. These results suggest that ambient mRNA correction effectively reduced the confounding influence of Ig genes, thereby revealing a more relevant immune signature in T cell subpopulations during acute dengue infection. We noted that the choice of ambient mRNA correction tools, such as DecontX [19], FastCAR [15], and scAR [37], may yield variable results in terms of adjusted expression valves. This variability may influence the outcomes of downstream analyses, including differential gene expression (e.g., log2 fold-change values) and pathway enrichment

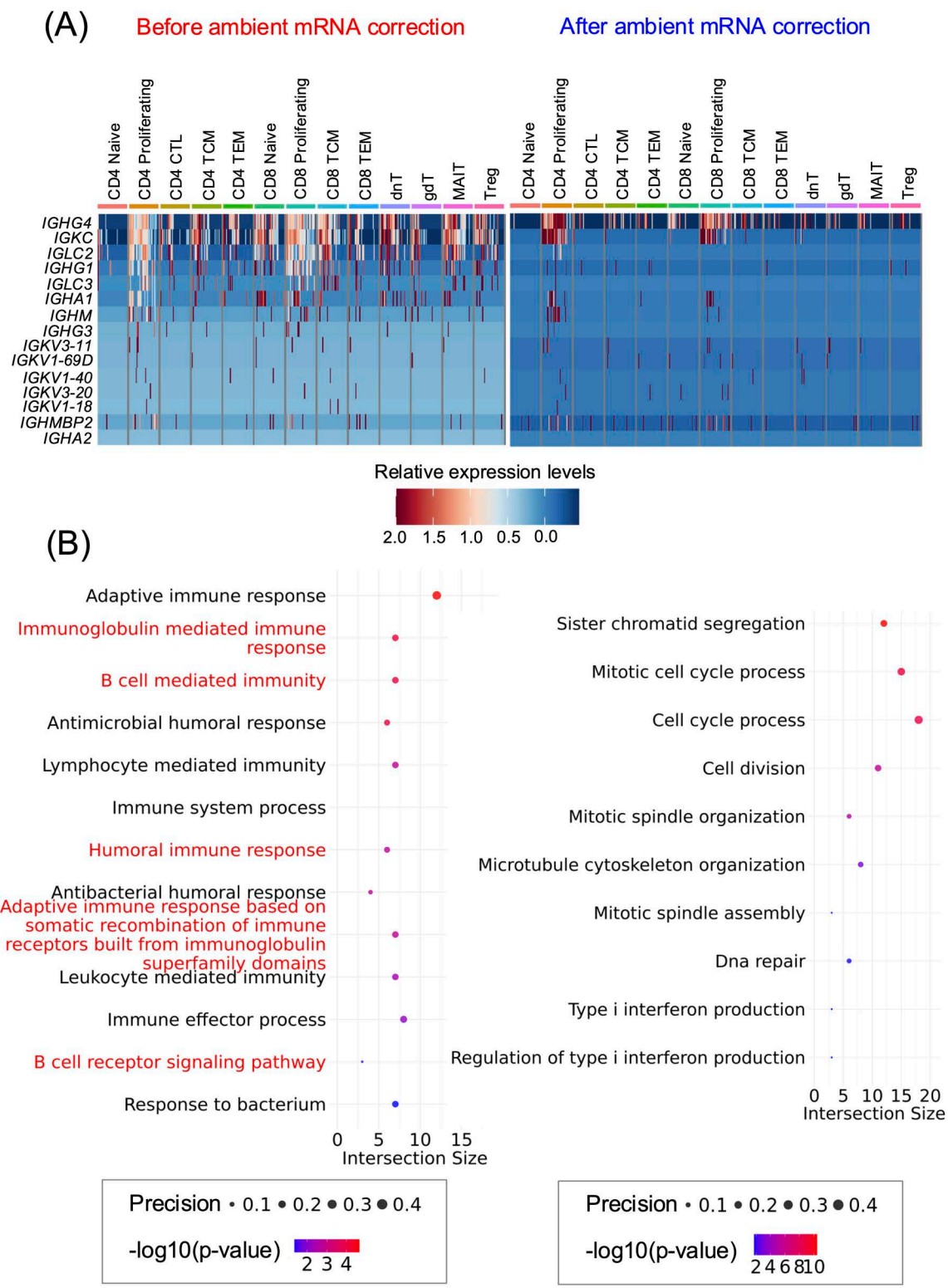

**Fig 3. Enhancement of T cell-specific DEGs and biological pathways after ambient mRNA contamination correction.** (A). Heatmap showing the relative transcription levels of Ig genes among the top 20 DEGs ranked by average log2 fold-change (avg_log2FC) in each biological condition (acute, convalescent, or healthy control) compared to the other two, across T cell subpopulations annotated using Azimuth, both before (left panel) and after

(right panel) ambient mRNA correction with CellBender (see S14 Fig for SoupX in S1 File). (B). The Gene Ontology Biological Processes (GO:BPs) of the DEGs from (A) before (left panel) and after (right panel) ambient mRNA correction with CellBender. Pathways labeled in red are B cell-specific pathways that have been previously reported in dengue infection study [13] (see S16 Fig for SoupX in S1 File).

analyses (e.g., p-values or significant pathways), as different tools make different assumptions and use different approaches for evaluating and removing ambient mRNA contamination.

## Improving differential gene expression and biological pathway enrichment in B cell subpopulations after ambient mRNA correction

We then asked whether the improvement in DEG identification and pathway enrichment after ambient mRNA correction could be seen in independent dataset and/or different cell types. Here, we obtained the scRNA-seq data of forty-two samples derived from human fetal liver tissues [21]. After data integration with no batch effects (S17 Fig in S1 File), hemoglobin (Hb) genes were identified as potential ambient mRNA contaminants, ranking among the top 5 genes predicted by SoupX (S14 Table). Consistently, we observed a large number of cells expressing Hb genes (S18 Fig in S1 File), which are known markers of erythroid (red blood cell) lineage, indicating contamination from Hb transcripts. We employed CellBender [17] and SoupX [20], using Hb genes as predefined potential ambient mRNA contamination gene list to estimate the contamination fractions in individual cells. Following the same analytic pipeline described above, we extracted B cell subpopulations as annotated by Azimuth, excluding non-B cells based on the annotation from the original study [21]. Similar to T cell subsets, the number of annotated B cell subpopulations remained relatively comparable before and after ambient mRNA correction (S9 Table and S19 Fig in S1 File). Here, we focused on B cell subsets because the contamination was primarily driven by Hb genes, which are markers of the erythroid lineage. Since B cells and erythroid cells originate from distinct progenitor lineages, this allowed us to assess the impact of ambient mRNA contamination from an unrelated cell type and evaluate the effectiveness of the correction.

Before ambient mRNA correction, nine Hb genes, namely *HBQ1, HBM, HBG1, HBG2, HBA1, HBA2, HBZ, HBB*, and *HBD*, were identified as DEGs across all annotated B cell subpopulations (Fig 4A; left panel and S20 Fig in S1 File; upper panel). After correction, Hb genes remained detectable but exhibited lower expression levels and were less widespread across B cell subsets (as shown using the same scale) in both CellBender (Fig 4A) and SoupX (S20 Fig in S1 File) results, with a more pronounced reduction in SoupX (S20 Fig in S1 File). Following correction with CellBender, eight of previously identified Hb genes remained among the DEGs, except for *HBQ1* (S20 Fig in S1 File; middle panel). On the contrary, after correction with SoupX, where Hb genes were predefined as potential ambient mRNAs, only three Hb genes (*HBQ1*, *HBA1*, and *HBD*) were identified among the DEGs (S20 Fig in S1 File; lower panel).

We further investigated the impact of ambient mRNA contamination on pathway analyses, focusing on GO:BPs derived from the top 2000 DEGs ranked by average log2 fold-change (avg_log2FC) in pro-B cells. Similar to PBMC-derived T cell subpopulations, we observed that biological pathways driven by Hb genes were uniquely identified before ambient mRNA correction but were absent after correction with both CellBender (Fig 4B and S15 Table) and SoupX (S22 Fig in S1 File and S16 Table). These pathways include GO:0007599 (hemostasis), GO:0050817 (coagulation), GO:0042743 (hydrogen peroxide metabolic process), GO:0030168 (platelet activation), GO:0015670 (carbon dioxide transport), and GO:0070527 (platelet aggregation) (Fig 4B, S22 Fig in S1 File, and S15-S16 Tables). The presence of these pathways suggests that ambient Hb gene contamination can lead to misleading enrichment of erythroid- and platelet-associated processes, which are biologically unrelated to pro-B cells. Notably, after ambient mRNA removal, these pathways were no longer detected (Fig 4B, S22 Fig in S1 File, and S15-S18 Tables), supporting the effectiveness of ambient contamination correction in eliminating improper signals and refining biological pathway interpretation. Pro-B cells are lineage-committed precursors

(A)

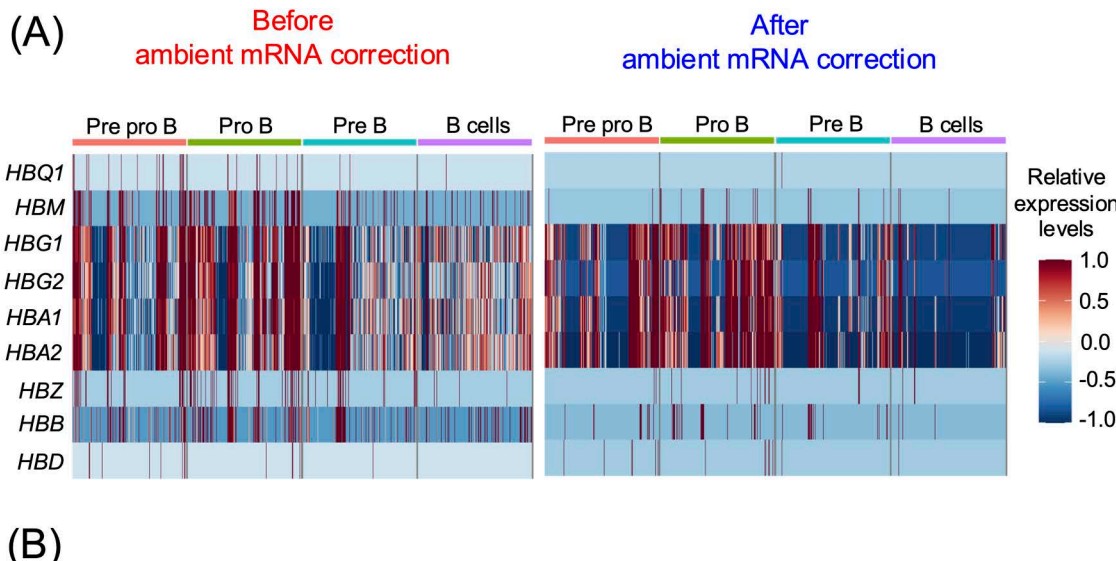

(B)

Before ambient mRNA correction

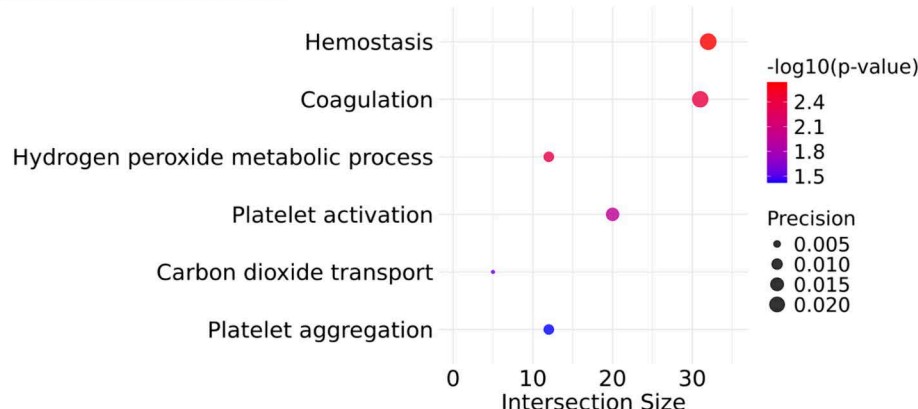

After ambient mRNA correction

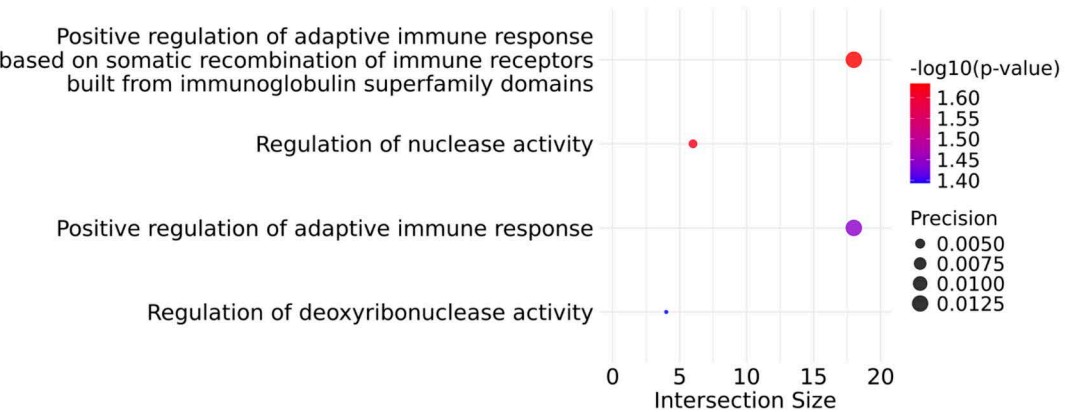

**Fig 4. Enhancement of B cell-specific DEGs and biological pathways after ambient mRNA contamination correction.** (A). Heatmap demonstrating the relative expression levels of Hb genes among the DEGs of each B cell subpopulation, as compared to the rest, comparing between before (left panel) and after (right panel) ambient mRNA correction with CellBender (see S21 Fig for SoupX in S1 File). B cell subpopulations were annotated

using Azimuth, excluding non-B cells based on the annotation from the original study [21]. (B). The Gene Ontology Biological Processes (GO:BPs) of the DEGs from top 2000 DEGs ranked by average log2 fold-change (avg_log2FC) in pro-B cells before (upper panel) and after (lower panel) ambient mRNA correction with CellBender (see S21 and S22 Figs for SoupX in S1 File).

of mature B cells, in which V(D)J recombination occurs to generate a functional B cell receptor, followed by positive selection. Consequently, pro-B cells were enriched in genes associated with recombination and DNA repair/stability (e.g., *HMGB1*, *PCNA*, *DDX11*, and *MSH2*), as well as genes involved in proliferation and cell survival (*AKT1*, *PDE3A*, and *TNFSF13B*). Importantly, these genes were observed in biological pathways that emerged only after correction in both CellBender (Fig 4B and S17 Table) and SoupX (S22 Fig in S1 File and and S18 Table), further supporting that ambient mRNA removal improves the accuracy of downstream functional analyses.

To further validate our findings, we leveraged a publicly available cross-species dataset from 10x Genomics (10-k-1-1-mixture-of-human-hek-293-t-and-mouse-nih-3-t-3-cells-3-v-3-1-3-1-standard-6-0-0, retrieved on 20 June 2025), in which human cells were intentionally contaminated with mouse transcripts, providing a well-defined "ground-truth" for cross-species ambient mRNA contamination (S23 Fig in S1 File). Similarly to observations in our T and B cell subpopulations, the contamination of mouse transcripts led to numerous false-positive DEGs and contributed to the presence of mouse transcripts in pathway enrichment analyses when no correction was applied (S24-S25 Figs in S1 File). After ambient mRNA correction, the cross-species contamination signals were remarkably reduced, as evidenced by using the same scales before and after correction (S24 Fig in S1 File), and no mouse genes were present in the significant biological pathways (S19 Table). Additionally, one GO:BP term (GO:0002181, cytoplasmic translation) was shared between before and after contamination correction. Prior to correction, genes in this GO:BP term were exclusively (100%) mouse genes, whereas after correction, this pathway was composed of human genes (S25 Fig in S1 File).

Taken together, our study highlights the practical applications of mitigating ambient mRNA contamination, including improved DEG identification and enhanced specificity of cell type-specific gene expression profiles and biological pathways in biological scRNA-seq datasets. Systematic studies using simulated or controlled experimental settings – for instance, those mimicking ambient contamination arising from cell death (e.g., abnormally elevated housekeeping, ribosomal, or mitochondrial genes), or overloaded droplets – represent a promising direction for further research to investigate the impact of ambient mRNA on the downstream analyses.

## Conclusion

Our study demonstrates the substantial improvements in data quality and biological insight achieved by addressing ambient mRNA contamination in single-cell transcriptome data. Specifically, we have shown that correcting ambient contamination enhances the identification of differentially expressed genes and refines biological pathway enrichment analyses, leading to more accurate interpretations of cell type-related functions. Our work emphasises the critical importance of appropriate ambient mRNA contamination correction in scRNA-seq preprocessing to enhance the robustness of biological interpretation within complex biological systems in single-cell RNA-seq datasets.

## Supporting information

**S1 Table. Know canonical markers.**
(XLSX)

**S2 Table. The total numbers of DEGs.**
(XLSX)

**S3 Table.  Average expression levels and percentages of cells expressing known canonical marker genes across annotated cell types, related to Fig 1B (dengue infection datasets).**
(XLSX)

**S4 Table.  Average expression levels and percentages of cells expressing known canonical marker genes across annotated cell types, related to Fig 1B (dengue infection datasets).**
(XLSX)

**S5 Table.  Top 50 potential ambient mRNA genes predicted by SoupX (dengue infection-dataset).**
(XLSX)

**S6 Table.  Average expression levels of known canonical marker genes across annotated cell types before and after correction, related to Fig 2 (dengue infection datasets).**
(XLSX)

**S7 Table.   Average expression levels of B cell-related genes across B and Non-B cells before and after correction, related to Fig 2 (dengue infection datasets).**
(XLSX)

**S8 Table.  Number and percentage of cells expressing Ig genes.**
(XLSX)

**S9 Table.   Numbers of annotated T and B cell subpopulations before and after correction. Related to Figs 3–4.**
(XLSX)

**S10 Table.   Average expression levels of immunoglobulin genes across T cell suppopulations before and after correction, related to Fig 3 (dengue infection datasets).**
(XLSX)

**S11 Table.  Significant 17 GO:BPs in T subsets before correction (dengue infection-dataset).**
(XLSX)

**S12 Table.  Significant 191 GO:BPs in T subsets CellBender correction (dengue infection-dataset).**
(XLSX)

**S13 Table.  Significant 46 GO:BPs in T subsets SoupX correction (dengue infection-dataset).**
(XLSX)

**S14 Table.  Top 50 potential ambient mRNA genes predicted by SoupX (fetal liver tissue-dataset).**
(XLSX)

**S15 Table.  Significant 279 unique GO:BPs in Pro-B cells before correction, compared to CellBender (fetal liver tissue-dataset).**
(XLSX)

**S16 Table.  Significant 365 unique GO:BPs in Pro-B cells before correction, compared to SoupX (fetal liver tissue-dataset).**
(XLSX)

**S17 Table.  Significant 60 unique GO:BPs in Pro-B cells CellBender correction (fetal liver tissue-dataset).**
(XLSX)

**S18 Table. Significant 47 unique GO:BPs in Pro-B cells SoupX correction (fetal liver tissue-dataset).** (XLSX)

**S19 Table. Significant 82 GO BPs in a species-mixing dataset (10x Genomics) after SoupX correction.** (XLSX)

**S1 File. Supporting Figs S1–S25.** (PDF)

## Acknowledgments

This research utilised Queen Mary's Apocrita HPC facility, supported by QMUL Research-IT (http://doi.org/10.5281/zenodo.438045). We acknowledge the ITS Research Team at QMUL for their support. Resources for data processing were also provided by Mahidol University and the Office of the Ministry of Higher Education, Science, Research and Innovation under the Reinventing University project: the Center of Excellence in AI-Based Medical Diagnosis (AI-MD) sub-project. We thank Sarintip Nguantad for running CellBender to preprocess single-cell RNA-seq data.

## Author contributions

**Conceptualization:** Jantarika Kumar Arora, Varodom Charoensawan.

**Data curation:** Jantarika Kumar Arora.

**Formal analysis:** Jantarika Kumar Arora.

**Funding acquisition:** Jantarika Kumar Arora, Louisa K. James, Varodom Charoensawan.

**Investigation:** Jantarika Kumar Arora, Louisa K. James, Varodom Charoensawan.

**Methodology:** Jantarika Kumar Arora, Varodom Charoensawan.

**Resources:** Louisa K. James.

**Software:** Jantarika Kumar Arora.

**Supervision:** Jantarika Kumar Arora, Louisa K. James, Varodom Charoensawan.

**Visualization:** Jantarika Kumar Arora.

**Writing – original draft:** Jantarika Kumar Arora, Varodom Charoensawan.

**Writing – review & editing:** Jantarika Kumar Arora, Louisa K. James, Varodom Charoensawan.

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
