## [Decision Letter · Decision Letter 0]

4 Sep 2024

PONE-D-24-33285Understanding and mitigating the impact of ambient mRNA contamination in single-cell RNA-sequencing analysisPLOS ONE

Dear Dr. Kumar Arora,

Thank you for submitting your manuscript to PLOS ONE. After careful consideration, we feel that it has merit but does not fully meet PLOS ONE’s publication criteria as it currently stands. Therefore, we invite you to submit a revised version of the manuscript that addresses the points raised during the review process.

We look forward to receiving your revised manuscript.

Kind regards,

Wan-Tien Chiang

Academic Editor

PLOS ONE

Journal Requirements:

"This project is funded by the mid-career researcher grant from National Research Council of Thailand (NRCT) and Mahidol University (N42A670557) through VC."

3. We noted in your submission details that a portion of your manuscript may have been presented or published elsewhere. [Single-cell RNA-seq datasets of dengue patients and one healthy donor were available through the ArrayExpress repository under accession number E-MTAB-9467. Additionally, one healthy sample was obtained from the 10x Genomics website (4k PBMCs from a healthy donor, Single Cell Gene Expression Dataset by Cell Ranger 2.1.0, 10x Genomics, 2017, November 8, https://support.10xgenomics.com/single-cell-gene-expression/datasets/2.1.0/pbmc4k). The raw and filtered gene-barcode matrices of the ten samples were obtained from: https://data.mendeley.com/datasets/6ry3x7r8hf/3. The raw and filtered gene-barcode matrices of forty single-cell experiments from human fetal liver tissues were deposited in the GigaDB repository, available at: http://gigadb.org/dataset/100836#.  ] Please clarify whether this [conference proceeding or publication] was peer-reviewed and formally published. If this work was previously peer-reviewed and published, in the cover letter please provide the reason that this work does not constitute dual publication and should be included in the current manuscript.

Reviewers' comments:

Reviewer's Responses to Questions

**Comments to the Author**

1. Is the manuscript technically sound, and do the data support the conclusions?

Reviewer #1: Partly

Reviewer #2: Partly

2. Has the statistical analysis been performed appropriately and rigorously? 

Reviewer #1: I Don't Know

Reviewer #2: No

3. Have the authors made all data underlying the findings in their manuscript fully available?

Reviewer #1: Yes

Reviewer #2: Yes

4. Is the manuscript presented in an intelligible fashion and written in standard English?

Reviewer #1: Yes

Reviewer #2: Yes

5. Review Comments to the Author

Reviewer #1: The authors present a study of the effects of removal of cell-free or "ambient" RNA on downstream interpretations of scRNA-seq data. The authors used publicly-available datasets for this study. This topic is an interesting one for the field, because removal of cell-free RNA has not yet become a ubiquitous part of scRNA-seq data analysis, and it is important for studies like this to demonstrate rigorously whether or not removal of cell-free RNA impacts biological findings in an important and quantitative way.

While the topic and stated aims of the study are of great interest, the analyses as presented may not go as far as I would personally like to see. I have several recommendations to strengthen the study.

Major concerns:

1. The major claim about the usefulness of removing ambient RNA is that it enhances "the accuracy of computational analyses and the robustness of data interpretation in scRNA-seq datasets." While I do agree with this statement, the results in the text focus on (1) increased log2-fold-changes of known marker genes, and (2) "notable improvement in clustering efficiency". (1) has been demonstrated by previous authors as cited in this manuscript. See the SoupX paper [15] Figure 3 for an analysis of PBMCs, which is quite similar to Figures 2 and 3 in this manuscript. And see the SoupX paper [15] Figure 4 for an analysis of kidney data, specifically showing the removal of the hemoglobin gene HBB, which is quite similar to Figure 4 in this manuscript. That is not to say that there is nothing novel here, but I am not sure that log-fold-changes of marker genes goes far enough to demonstrate the utility of ambient RNA removal.

2. Major claim (2) above – “notable improvement in clustering efficiency” – may be a bit problematic. How do the authors quantify improvements in clustering efficiency? What does clustering “efficiency” mean? In Figure 4 for example, I see UMAPs before and after correction. And I see a plot of PCs before and after correction. The problem with the UMAP is that it is qualitative. The problem with the PC plot is that I am not sure whether the movement of dots on this plot is meaningful. How meaningful? How can this be quantified?

3. The authors state in the introduction that “the effects of ambient mRNA correction on certain downstream analyses … remains largely unclear.” While I agree, the authors should probably note that other studies have been carried out which focus largely on this question, including PMID: 36240767, but there may be others.

4. The authors focused the study solely on the SoupX method. While it makes sense to limit the scope of work, it is unclear whether the conclusions would be the same if DecontX or CellBender were used. (See suggestions below.)

Minor concerns:

1. The dataset integration in the UMAP in Figure S9 looks potentially problematic. Was the data integrated by the authors? How? Is that the statement in the Methods about “processed as described in the original article”? I am not sure this would give me enough information to reproduce the analysis here. What does that UMAP look like if cells are colored by “batch”?

2. CD79A is listed in the text as a B cell marker, but shows up in Table S1 as a plasma cell marker.

3. It is unclear to me whether it is sound to run SoupX with a manually-provided set of hemoglobin genes, and then look at the output performance on hemoglobin genes to show that SoupX worked well. Namely “virtually no Hb genes were detected after the ambient mRNA correction.” Is this a good thing? How do we know that SoupX is not overcorrecting? How do we know if SoupX worked for other genes which were not provided on the input “potential contamination gene list”?

4. TUBB and TUBA1B are referred to as “adhesion molecules”, which I am not sure is accurate.

5. Section 3.4 begins “We then asked if the improvement on cell type identification and DEG analyses as a result of ambient mRNA correction can be seen in other independent dataset…” Did the authors demonstrate improvement in cell type identification? I would be very interested to see this, but I am not sure this was clearly demonstrated here. Something like scPred (PMID: 31829268) or another automatic-cell-type-annotation tool could potentially be used to annotate the datasets before and after ambient RNA removal. Then the performance enhancement could be quantified.

6. How are there so many B cell populations in Fig. 4 and Fig. S11? There appear to be 12 B cell clusters after correction. This seems like a large number of clusters of B cells. Are these biological meaningful clusters, and how can a reader be convinced of that? Or are we over-clustering here? What does this UMAP look like if cells are colored by “batch”?

7. I was confused about the point being made in Fig. 4. What is the relationship between the cells in “before correction” cluster 13 and “after correction” clusters 10 and 11? Are they the same cells? In general, how do the authors identify which “before correction” cluster corresponds to which “after correction” cluster (especially as there are different numbers of clusters)?

8. What does Fig. 4C look like “before correction”, if you use the same “after correction” cluster labels for cluster 10 and 11? Are the histone-related gene markers something that is only seen after ambient RNA correction?

Suggestions:

This study would be much more impactful if it either (1) went further with analyses of downstream effects, and/or (2) compared multiple methods of ambient RNA removal, rather than SoupX alone.

Recommendations:

I would really like to see a study like this published, and I think the authors have followed an interesting line of work. However, I have some concerns about whether the current analyses as presented go farther than what is shown in the literature. In particular, there is work that seems comparable in Figures 3 and 4 of [15], or Figure 2 and 3 of [14], or Figure 4 and 5 of [16]. I recommend this article should be published with some subset of the above-suggested revisions.

Reviewer #2: Please see the attached comments.

6. PLOS authors have the option to publish the peer review history of their article (what does this mean? ). If published, this will include your full peer review and any attached files.

**Do you want your identity to be public for this peer review?** For information about this choice, including consent withdrawal, please see our Privacy Policy .

Reviewer #1: No

Reviewer #2: No

---

## [Author Response · Author response to Decision Letter 1]

27 Feb 2025

Dear Dr. Wan-Tien Chaing, PLOS ONE Editorial Team, and Reviewers,

We would like to thank you for the opportunity to improve our manuscript and for the insightful comments that have helped enhance the quality of our study. We have carefully addressed and incorporated the suggestions into our revised manuscript. Please find point-by-point responses and specific details of the changes made in the“Response to Reviewers” document.

Best regards,

Jantraika Kumar Arora (on behalf of the authors)

---

## [Decision Letter · Decision Letter 1]

20 May 2025

PONE-D-24-33285R1Understanding and mitigating the impact of ambient mRNA contamination in single-cell RNA-sequencing analysisPLOS ONE

Dear Dr. Kumar Arora,

Thank you for submitting your manuscript to PLOS ONE. After careful consideration, we feel that it has merit but does not fully meet PLOS ONE’s publication criteria as it currently stands. Therefore, we invite you to submit a revised version of the manuscript that addresses the points raised during the review process.

We look forward to receiving your revised manuscript.

Kind regards,

Wan-Tien Chiang

Academic Editor

PLOS ONE

Journal Requirements:

Reviewers' comments:

Reviewer's Responses to Questions

**Comments to the Author**

1. If the authors have adequately addressed your comments raised in a previous round of review and you feel that this manuscript is now acceptable for publication, you may indicate that here to bypass the “Comments to the Author” section, enter your conflict of interest statement in the “Confidential to Editor” section, and submit your "Accept" recommendation.

Reviewer #3: All comments have been addressed

Reviewer #4: (No Response)

2. Is the manuscript technically sound, and do the data support the conclusions?

Reviewer #3: Yes

Reviewer #4: Partly

3. Has the statistical analysis been performed appropriately and rigorously? 

Reviewer #3: I Don't Know

Reviewer #4: (No Response)

4. Have the authors made all data underlying the findings in their manuscript fully available?

Reviewer #3: Yes

Reviewer #4: (No Response)

5. Is the manuscript presented in an intelligible fashion and written in standard English?

Reviewer #3: Yes

Reviewer #4: Yes

6. Review Comments to the Author

Reviewer #3: 1. For introduction part: regrading impact of ambient mRNA contamination in single-cell RNA-sequencing analysis on differential gene expression and biological pathway enrichment analyses, can authors explain the differences between real biological findings and artifacts induced by ambient mRNA contamination?

2. Methods/results: authors may use simulated dataset to address impact of ambient mRNA contamination in single-cell RNA-sequencing analysis on differential gene expression and biological pathway enrichment analyses in following different simulated matrix if ambient mRNA contamination are due to following reasons:

1) Cell death or damage: background gene expression signals; Marker genes appearing unexpectedly in non-target cells (false positives); Abnormally elevated housekeeping, ribosomal, and mitochondrial genes

2) Cross-contamination in library preparation: Low-level expression of highly cell-type-specific genes in unrelated cell populations

3) Improper cell concentration: Overloaded cells result in multiple cells per droplet (doublets); Mixed expression patterns from multiple cell types (e.g., T cell and monocyte markers simultaneously detected)

3. Methods/results: for determine impact of ambient mRNA contamination in single-cell RNA-sequencing analysis on differential gene expression and biological pathway enrichment analyses, authors may need explain quantitative standards, not just using qualitative methods such as "increase/decrease"

4. Discussion: authors may need explain potential benchmakt effects for many related softwares such as DecontX, scAR

Reviewer #4: The authors applied two computational methods (SoupX and CellBender) to estimate and remove ambient mRNAs in droplet-based scRNA-seq datasets and observed reduced false positive differential expressed genes (DEGs) on 2 cell types of interests and subsequently improved biological pathways. The results in this manuscript show dataset dependent performance improvements as well as some level of persisting artifacts after using decontamination methods. This would serve as a useful reference for the field to draw analysis conclusions with caution and scrutiny on scRNAseq analysis.

I appreciate the comments from Reviewer 1 and Reviewer 2 and authors’ additional analysis and improved clarity. I have a few suggestions that I believe would be necessary to further improve the manuscript:

1. Authors should report cell-level as well as dataset level contamination percentage estimated by both methods (SoupX and CellBender).

As pointed out by reviewer 2, a simulation study could further illustrate how contamination level would affect downstream analysis. Although simulation itself could potentially require the work that’s beyond the scope of this manuscript, a deeper insight could be provided by reporting the estimated contamination levels by both methods and commenting on their associations with dataset dependent improvements.

2. 2000 DEGs are a lot for a subpopulation – i.e. a highly functionally specific cell type (“top 2000 DEGs in pro-B cells” Page 6 Line 156). And this result is even more concerning when they are only the positive set of the DEGs (“the positive markers were retained (Fig 3A and 4A).” Page 6 Line 148) – implying the whole DEG set including both the positively and negatively differentially expressed gene pool is even larger.

Can the authors explain whether this is biologically plausible? In addition, authors should report the number of genes used for performing DE analysis as well as the resulting number of DEGs.

3. Fig S9 and S19 provided relative cell-type composition information but lacked absolute value information – the number of cells in each type before and after ambient mRNA correction. Cell-type assignment could change after correction, and it would be informative to show magnitude of such changes for both methods given two very different methodological approaches between SoupX and CellBender.

7. PLOS authors have the option to publish the peer review history of their article (what does this mean? ). If published, this will include your full peer review and any attached files.

**Do you want your identity to be public for this peer review?** For information about this choice, including consent withdrawal, please see our Privacy Policy .

Reviewer #3: No

Reviewer #4: No

---

## [Author Response · Author response to Decision Letter 2]

3 Jul 2025

We would like to thank you for the opportunity to revise our manuscript and for the insightful comments that have significantly improved the quality of our study. We have carefully addressed and incorporated the suggestions into the revised manuscript. Please find our point-by-point responses and specific details of the changes made in the “Response to Reviewers” document.

---

## [Decision Letter · Decision Letter 2]

29 Jul 2025

PONE-D-24-33285R2Understanding and mitigating the impact of ambient mRNA contamination in single-cell RNA-sequencing analysisPLOS ONE

Dear Dr. Kumar Arora,

Thank you for submitting your manuscript to PLOS ONE. After careful consideration, we feel that it has merit but does not fully meet PLOS ONE’s publication criteria as it currently stands. Therefore, we invite you to submit a revised version of the manuscript that addresses the points raised during the review process.

We look forward to receiving your revised manuscript.

Kind regards,

Wan-Tien Chiang

Academic Editor

PLOS ONE

Journal Requirements:

Reviewers' comments:

Reviewer's Responses to Questions

**Comments to the Author**

1. If the authors have adequately addressed your comments raised in a previous round of review and you feel that this manuscript is now acceptable for publication, you may indicate that here to bypass the “Comments to the Author” section, enter your conflict of interest statement in the “Confidential to Editor” section, and submit your "Accept" recommendation.

Reviewer #3: All comments have been addressed

Reviewer #4: (No Response)

2. Is the manuscript technically sound, and do the data support the conclusions?

Reviewer #3: Yes

Reviewer #4: (No Response)

3. Has the statistical analysis been performed appropriately and rigorously? 

Reviewer #3: Yes

Reviewer #4: (No Response)

4. Have the authors made all data underlying the findings in their manuscript fully available?

Reviewer #3: Yes

Reviewer #4: (No Response)

5. Is the manuscript presented in an intelligible fashion and written in standard English?

Reviewer #3: (No Response)

Reviewer #4: (No Response)

6. Review Comments to the Author

Reviewer #3: Last question remarks for this manuscript were mostly addressed:

1. Introduction: Add a new example;

2. Methods: include a new ambient mRNA correction method;

3. Methods/Results: statistical methods were used in celltype annotation/DE analysis/pathway enrichment;

4. Add a new validation dataset.

Reviewer #4: Thanks to the authors addressing clearly to my comments.

There is a minor clarification I would request authors to state in the manuscript about log-fold change threshold:

It was stated DE analysis's criterium on log-fold change is 0.1, "a minimum log2 fold-change threshold of 0.1 " (Page 6 Line 157). for the purpose of examining false positive DE genes in the context of this manuscript is a okay. However this should not be used in general to draw biological conclusions upon as log2 ratio being 0.1 means the expression ratio of the 2 compared groups is 1.0717, i.e. one group has about 7% higher expression than the other which is extremely low.

Hence authors should state in the manuscript that this threshold 0.1 was chosen to have a larger gene pool to show ambient RNAs' bias effect on DE analysis rather as the normal standard for DE analysis.

7. PLOS authors have the option to publish the peer review history of their article (what does this mean? ). If published, this will include your full peer review and any attached files.

**Do you want your identity to be public for this peer review?** For information about this choice, including consent withdrawal, please see our Privacy Policy .

Reviewer #3: No

Reviewer #4: No

---

## [Author Response · Author response to Decision Letter 3]

2 Aug 2025

We greatly appreciate the reviewers’ thoughtful and insightful feedback. We have carefully addressed all comments and incorporated the suggested revisions into the manuscript. Please find our point-by-point responses and details of the changes in the “Response to Reviewers” document.

Sincerely,

Jantarika Kumar Arora (on behalf of the authors)

---

## [Editor Report · Decision Letter 3]

31 Aug 2025

Understanding and mitigating the impact of ambient mRNA contamination in single-cell RNA-sequencing analysis

PONE-D-24-33285R3

Dear Dr. Kumar Arora,

We’re pleased to inform you that your manuscript has been judged scientifically suitable for publication and will be formally accepted for publication once it meets all outstanding technical requirements.

Kind regards,

Wan-Tien Chiang

Academic Editor

PLOS ONE
---

## [Editor Report · Acceptance letter]

PONE-D-24-33285R3

PLOS ONE

Dear Dr. Kumar Arora,

I'm pleased to inform you that your manuscript has been deemed suitable for publication in PLOS ONE. Congratulations! Your manuscript is now being handed over to our production team.

Kind regards,

on behalf of

Dr. Wan-Tien Chiang

Academic Editor

PLOS ONE